# Blackleg Yield Losses and Interactions with Verticillium Stripe in Canola (*Brassica napus*) in Canada

**DOI:** 10.3390/plants12030434

**Published:** 2023-01-17

**Authors:** Yixiao Wang, Stephen E. Strelkov, Sheau-Fang Hwang

**Affiliations:** Department of Agricultural, Food and Nutritional Science, University of Alberta, Edmonton, AB T6G 2P5, Canada

**Keywords:** blackleg, *Brassica napus*, canola, interactions, Verticillium stripe, yield losses

## Abstract

Blackleg, caused by *Leptosphaeria maculans*, is an important disease of canola (*Brassica napus*). The pathogen can attack stems, leaves and pods, but basal stem cankers are most damaging and can result in significant yield losses. In Canada, Verticillium stripe (*Verticillium longisporum*) has recently emerged as another disease threat to canola. Symptoms of Verticillium stripe can resemble those of blackleg, and the two diseases may occur together. The effect of blackleg on yield was explored in field experiments with two canola hybrids and by evaluating a wider variety of hybrids in commercial crops in central Alberta, Canada. The impact on yield of *L. maculans*/*V. longisporum* interactions was also assessed under field and greenhouse conditions. In most hybrids, the relationship between blackleg severity and yield components was best explained by second-degree quadratic equations, although a linear relationship was found for one variety sampled in commercial fields. When *L. maculans* was co-inoculated with *V. longisporum*, blackleg severity and yield losses increased. In some cases, Verticillium stripe caused greater yield losses than blackleg. The results suggest that the interaction between *L. maculans*/*V. longisporum* may cause more severe losses in canola, highlighting the need for proactive disease management strategies.

## 1. Introduction

Blackleg, caused by the fungus *Leptosphaeria maculans* (Desm.) Ces. et De Not. (anamorph: *Phoma lingam* (Tode) Desm.), is an important disease of canola (oilseed rape; *Brassica napus* L.) in Europe, Australia, and Canada [1,2,3]. The fungus can attack the stems, leaves and pods of *B. napus*, but the formation of basal stem cankers is most damaging [4]. These cankers appear dry and sunken and may be dotted with black pycnidia, the asexual fruiting bodies of *L. maculans*. In cross-section, a dark discoloration of the vascular tissues is visible in infected canola stems, which may help to diagnose the disease. In recent surveys of western Canadian canola crops, blackleg was found to occur at an average incidence of 12.5% and an average prevalence of 80% [5,6,7]. The deployment of blackleg-resistant cultivars is the most effective strategy for blackleg management [8]. However, the erosion or loss of resistance has been reported in many canola cultivars, reflecting the emergence of virulent isolates of *L. maculans* [9,10].

Severe epidemics of blackleg can result in yield losses of 30–50% [11,12,13]. In experiments conducted in western Canada, Hwang et al. [14] found a negative linear relationship between blackleg severity and yield loss in a susceptible canola cultivar, ‘Westar’, with a 17.2% reduction in yield for each unit increase in disease severity, as assessed on a 0–5 scale. More recently, Wang et al. [15] examined blackleg severity-yield loss relationships in moderately resistant canola hybrids and reported quadratic relationships between blackleg severity and yield losses. A small increase in yield was found under very mild infection in plants rated as ‘1’ on the 0–5 scale, which was followed by large yield decreases at blackleg severities of ≥2. At a severity rating of ‘5’, the percentage yield loss was as high as 99% [15]. In canola hybrids classified as blackleg resistant, which represent most of the cultivars available in the Canadian market [16], the relationship between blackleg severity and yield loss has not been explored.

Over the last few years, another disease, Verticillium stripe caused by *Verticillium longisporum* (C. Stark) Karapapa, Bainbridge and Heale, has also emerged as a potential threat to canola in western Canada. Initially identified in Manitoba in 2014, the presence of *V. longisporum* was soon confirmed across much of the country, including British Columbia, Alberta, Saskatchewan, Ontario, and Quebec [17]. By 2021, symptoms of Verticillium stripe were observed in nearly 3% and 30% of canola crops surveyed in Alberta [6] and Manitoba [7], respectively. Three samples collected in Saskatchewan also tested positive for the presence of *V. longisporum* [5]. Symptoms and signs of Verticillium stripe include necrosis and shredding of the stem tissue, along with the presence of pathogen microsclerotia [18]. Infection by *V. longisporum* also results in staining of the vascular tissues, visible in cross-sections of the stem, which can be confused with the symptoms caused by *L. maculans* described above. Yield losses associated with Verticillium stripe were between 10 to 50% in oilseed rape in Sweden [19].

Considering that both blackleg and Verticillium stripe now occur on canola in western Canada and given the generally greater familiarity of farmers and agronomists with the former [5,6,7], it is possible that some *V. longisporum* infections are being misdiagnosed as *L. maculans*. It is also likely that some canola plants are infected by both pathogens. In this context, it is important to assess if and how co-infection of canola affects yield losses caused by *V. longisporum* and *L. maculans*. Moreover, given the linear vs. quadratic relationships reported [14,15] in susceptible vs. moderately resistant canola cultivars, respectively, it is also important to clarify this relationship in hosts classified as resistant. The main objectives of this study were: (1) to establish the relationship between blackleg severity and the yield of blackleg resistant canola hybrids under experimental field conditions and in commercial crops in western Canada and (2) to examine possible interactions between blackleg and Verticillium stripe with respect to yields. In addition, specific symptoms were compared that could help non-expert personnel to more easily distinguish between blackleg and Verticillium stripe.

## 2. Results

### 2.1. Field Experiments for Blackleg Yield Losses

Mean blackleg disease severity on the non-inoculated canola hybrids ‘45H31’ and ‘CS2000’ was 0.1 and 0.2, respectively, in 2019, and 0.0 for both hybrids in 2020 (Figure 1). On the inoculated treatments, mean blackleg disease severity was 2.4 on ‘45H31’ and 1.4 on ‘CS2000’ in 2019, and 1.3 and 0.8, respectively, in 2020 (Figure 1). A paired t-test indicated that the mean disease severities among the inoculated cultivars were different at *p* < 0.01. In 2019, the seed yield of the non-inoculated canola ‘45H31’ and ‘CS2000’ was 3.47 and 2.72 t ha^−1^, respectively, while in the inoculated plots, the yield was 2.18 and 1.87 t ha^−1^, respectively (Table 1). In 2020, the seed yield of the non-inoculated canola ‘45H31’ and ‘CS2000’ was 1.05 and 0.88 t ha^−1^, respectively, while in the inoculated plots, yield was 1.00 and 0.82 t ha^−1^, respectively (Table 1).

Regression analysis indicated that pod number and seed yield declined with increasing blackleg severity. When averaged across two years for the canola hybrid ‘45H31’, the average seed yield and pod number (+/− SE) ranged from 5.30 g ± 1.28 g to 41.18 g ± 3.55 g seed per plant and from 44 ± 12 to 433 ± 79 pods per plant. The regression models were y = −1.767x^2^ − 65.456x + 419.093 (*p* < 0.01, R^2^ = 0.97) for pod number vs. disease severity (Figure 2a), and y = −0.2871x^2^ − 5.9911x + 42.24 (*p* < 0.01, R^2^ = 0.97) for seed yield vs. disease severity (Figure 2b). Average seed yield and pod number for the canola hybrid ‘CS2000’ ranged from 2.74 g ± 1.05 g to 27.77 g ± 6.37 g seed per plant and from 91 ± 35 to 493 ± 137 pods per plant. The regression models were y = 2.721x^2^ − 92.035x + 454.402 (*p* = 0.08, R^2^ = 0.69) for pod number vs. disease severity (Figure 2a), and y = −0.0652x^2^ − 4.4641x + 24.836 (*p* = 0.08, R^2^ = 0.69) for seed yield vs. disease severity (Figure 2b).

The regression models for percentage yield losses vs. disease severity were y = 0.698x^2^ + 14.5456x − 2.5786 (*p* < 0.01, R^2^ = 0.97) for ‘45H31’, and y = 0.3114x^2^ + 21.4063x − 19.0623 (*p* = 0.08, R^2^ = 0.69) for ‘CS2000’ (Figure 3). In ‘CS2000’, plants with a blackleg severity of 0 had a slightly lower yield than plants with a severity of 1. When disease severity was rated as 1, seed yield increased by 6.91 g. However, as disease severity increased further from 2 to 5, yields began to decrease. The percentage yield loss increased by 26.86%–86.87% in plants with disease severities of 2–5, relative to plants with disease severities of 0–1. In contrast, on ‘45H31’, seed yield begun to decrease at a disease severity rating of 1 and continued to decrease as severity increased to a rating of 5. However, a high-adjusted R^2^ value indicated the relationship between yield loss and blackleg severity still fit a second-degree quadratic equation better than a linear regression.

### 2.2. Blackleg Yield Losses in Commercial Crops

Symptoms of blackleg were identified in all nine canola crops sampled in central Alberta. Mean disease severities on the hybrids ‘DKTF 94CR’ and ‘75-42CR’ were 2.9 and 2.7, respectively, across all the crops. On ‘DKTF 94CR’, regression analysis indicated that the average seed yield ranged from 1.80 g ± 0.55 g to 17.69 g ± 3.71 g per plant, while the average pod number ranged from 76 ± 18 to 271 ± 61 pods per plant. The regression model was y = –8.7969 x^2^ + 20.969x + 194.61 (*p* = 0.22, R^2^ = 0.39) for pod number vs. disease severity (Figure 4a), and y = −0.2664x^2^ − 1.297x + 14.138 (*p* = 0.11, R² = 0.61) for seed yield vs. disease severity (Figure 4b). In the case of ‘75-42CR’, the average seed yield ranged from 0.80 g ± 0.27 g to 15.79 g ± 3.42 g per plant, while the average pod number ranged from 45 ± 18 to 276 ± 88 pods per plant. The regression model was y = –32.367x + 233.3 (*p* = 0.15, R^2^ = 0.31) for pod number vs. disease severity (Figure 4a), and y = −3.1563x + 15.911 (*p* < 0.01, R² = 0.96) for seed yield vs. disease severity (Figure 4b).

The regression models for percent yield loss vs. disease severity were y = 2.2678x^2^ + 11.04x − 20.339 (*p* = 0.11, R^2^ = 0.61) for ‘DKTF 94CR’ and y = 19.992x − 0.7785 (*p* < 0.01, R^2^ = 0.96) for ‘75-42CR’ (Figure 5). In the case of the hybrid ‘DKTF 94CR’, plants with a blackleg severity of 1 had a greater seed yield relative to plants with a rating of 0, but as severities increased from 2 to 5, yields began to decrease and the percentage yield loss increased from 26.4–84.7% relative to plants with disease severities of 0 or 1. In contrast, a linear relationship between disease severity and percentage yield loss was observed in ‘75-42CR’. For each unit increase in blackleg severity, estimated yield losses were approximately 20%.

### 2.3. Field Experiments for Blackleg and Verticillium Stripe Interactions

Mean blackleg disease severity ranged from 0.1 to 1.6 on ‘45H31’, and from 0.0 to 1.3 on ‘CS2000’, at the two sites over two years (Table 2). On ‘45H31’ in 2020, the most severe blackleg (1.3–1.6) at site 1 was observed in treatments inoculated with *V. longisporum* alone or with a 3:1 or 1:1 mix of *L. maculans* and *V. longisporum*; in the *L. maculans* alone treatment, the blackleg severity (1.0) was significantly lower than in the 1:1 mix of pathogens. At site 2 in 2020, the most severe blackleg (1.2–1.5) on ‘45H31’ developed following inoculation with the 3:1 and 1:1 mixes of *L. maculans* and *V. longisporum*, while the lowest disease (0.1) was observed on the control. The *L. maculans* alone and 1:3 mix of *L. maculans* and *V. longisporum*, and *V. longisporum* alone treatments developed intermediate blackleg severities (0.7–1.0). On ‘CS2000’ in 2020, all inoculated treatments developed blackleg severities ranging from 0.8 to 1.3 at the two sites, which was significantly greater than the severity (0.1) on the non-inoculated control (Table 2). In 2021 at site 1, the most severe blackleg (1.5) on ‘45H31’ developed on the *L. maculans* alone treatment, and the mildest blackleg was observed on the control (0.5) and 1:3 mix of *L. maculans* and *V. longisporum* (0.3) and *V. longisporum* alone (0.7) treatment; the disease severity on the other treatments was intermediate (Table 2). Similar trends were observed for ‘45H31’ at site 2 and ‘CS2000’ at sites 1 and 2 in 2021; the most severe blackleg developed on the *L. maculans* alone treatment, in which disease was generally higher than most other treatments, although the mildest symptoms did not always occur on the non-inoculated control (Table 2).

The mean Verticillium stripe severity ranged from 0.0 to 2.2 on the hybrid ‘45H31’ and from 0.0 to 2.0 on ‘CS2000’ at the two sites over two years (Table 3). At site 1 in 2020, the numerically most severe Verticillium stripe (0.5) on ‘45H31’ was observed in the *V. longisporum* alone treatment, although this was not significantly greater than the disease (0.4) that developed following inoculation with the 3:1 and 1:1 mixes of the pathogens. However, Verticillium stripe on the *V. longisporum* alone treatment was significantly more severe than on the control (0.0), 1:3 mix of *L. maculans* and *V. longisporum* (0.2), and *L. maculans* alone (0.3) treatments for ‘45H31’ at site 1 in 2020. At site 2 in 2020, there were no significant differences in Verticillium stripe severity on this hybrid (Table 3). In the case of ‘CS2000’ at both sites in 2020, the most severe Verticillium stripe (0.6–0.7) was observed in treatments inoculated with *V. longisporum*, while the mildest disease was found on the control (0.1–0.2) and *L. maculans* alone (0.2–0.3) treatments (Table 3). In 2021 on ‘45H31’ at site 1, the most severe Verticillium stripe (1.0–1.2) was observed on treatments inoculated with 3:1, 1:1, 1:3 mixes of *L. maculans* and *V. longisporum*, as well as on the *V. longisporum* alone treatment at site 1. The no inoculum control (0.5) and *L. maculans* alone (1.1) treatments were significantly different from other treatments. While the *L. maculans* alone treatment had a high numerical value (1.1) relative to all other inoculated treatments; the low standard deviation resulted in significant differences. At site 2 in 2021, the most severe Verticillium stripe was observed in the 1:3 mix of *L. maculans* and *V. longisporum* (1.8) and *V. longisporum* alone (1.9) treatments. The mildest Verticillium stripe was observed on the control (0.6) and *L. maculans* alone (1.1) treatments (Table 3). In the case of ‘CS2000’, the most severe Verticillium stripe was observed on the 1:3 mix of *L. maculans* and *V. longisporum* (1.2) and the *V. longisporum* alone (1.6) treatments. The mildest Verticillium stripe was observed on the control (0.2) and *L. maculans* alone (0.4) treatments at site 1 in 2021. At site 2, the control (0.0) and *L. maculans* alone (0.2) treatments had the lowest Verticillium stripe and was significantly different from other treatments (1.5–2.0). (Table 3).

The mean seed yield was similar in the two hybrids, ranging from 0.9 to 2.5 t/ha on ‘45H31’ and from 0.9 to 2.8 t/ha on ‘CS2000’ and was significantly greater in 2021 than in 2020 (Figure 6a,b). However, the mean seed yield was not significantly different among treatments for either hybrid in either year.

### 2.4. Comparison of Symptoms and Signs on Canola

While the symptoms and signs of blackleg and Verticillium stripe were superficially similar, they could readily be distinguished with careful examination, even when they occurred together. The microsclerotia of *V. longisporum* were much smaller than the pycnidia produced by *L. maculans* and were greyer in color (Figure 7a). Due to their larger size, individual pycnidia could be discerned more easily, and were generally more darkly pigmented than the microsclerotia. Moreover, while both *V. longisporum* and *L. maculans* caused a vascular discoloration visible in cross-sections of the crown or base of the stem, the staining associated with blackleg was darker (black) and more discrete than the grey, more diffuse staining resulting from Verticillium stripe (Figure 7b). Longitudinal sections of the stem further served to distinguish the two diseases. In the case of blackleg, the vascular discoloration was restricted to the lower stem, affecting the cortex and epidermis (Figure 8a); in the case of Verticillium stripe, symptoms extended up the stem, with a hollow, darker center (Figure 8b). In cases where the two pathogens occurred together, longitudinal sections revealed a hollow and darker center together with black discoloration of the cortex and epidermis (Figure 8c). Infection by *V. longisporum* also was usually associated with some shredding of the stem.

### 2.5. Greenhouse Experiments for Blackleg and Verticillium Stripe Interactions

Emergence ranged from 41.6 to 94.7% and from 51.9 to 95% in the canola hybrids ‘45H31’ and ‘CS2000’, respectively, 14 days after seeding in the greenhouse experiments (Table 4). For both hybrids, percent emergence was highest (~95%) in the control (non-inoculated) treatments. In the case of ‘CS2000’, the emergence in all of the treatments that received any inoculum (regardless of the ratio of *L. maculans* and *V. longisporum*) was similar (51.9–63.8%). In contrast, for ‘45H31’, the lowest emergence was observed in the *L. maculans* only treatment and in the 3:1 mix of *L. maculans* and *V. longisporum* (41.6–42.2%), followed by the 1:1 and 1:3 *L. maculans*/*V. longisporum* mixes and the *V. longisporum* only treatment (54.7–57.2%).

The mean blackleg severity ranged from 0.0 to 1.3 on ‘45H31’, with no blackleg detected (severity of 0.0) in either the no inoculum control or *V. longisporum* only treatment. On this hybrid, the greatest blackleg severity (1.3) was obtained with the 3:1 mix of *L. maculans* and *V. longisporum*, followed by the *L. maculans* only (0.9) and 1:3 *L. maculans*/*V. longisporum* (0.6) treatments (Table 4). On ‘CS2000’, blackleg severity ranged from 0.0 to 1.0, with no symptoms of the disease detected on the non-inoculum control or *V. longisporum* only treatment. The most severe blackleg on ‘CS2000’ was obtained with the 3:1 and 1:1 mixes of *L. maculans* and *V. longisporum*, as well as with the *L. maculans* only treatment (severities of 0.8 to 1.0) (Table 4). The mean Verticillium stripe severity ranged from 0.0 to 1.9 on both ‘45H31’ and ‘CS2000’, with no Verticillium stripe detected in the no inoculum control or *L. maculans* only treatment for either hybrid (Table 4). On ‘45H31’, the highest Verticillium stripe severity was observed in the *V. longisporum* only treatment, followed by intermediate severities (1.0 to 1.3) in the 3:1, 1:1 and 1:3 *L. maculans*/*V. longisporum* treatments. On ‘CS2000’, the most severe (1.6–1.9) Verticillium stripe was observed in any treatment that included *V. longisporum*, regardless of the ratio or whether or not *L. maculans* was included (although there seemed to be a numerical increase in severity as the proportion of *V. longisporum* increased) (Table 4).

Mean seed yield ranged from 1.5 g to 3.9 g per plant on ‘45H31’ and from 1.2 g to 2.3 g per plant on ‘CS2000’ (Table 4). For ‘45H31’, the lowest yields were observed in the non-inoculated control and *V. longisporum* only treatments, followed by the 1:3 mix of *L. maculans*/*V. longisporum*. The highest yields were obtained in the *L. maculans* only and 3:1 *L. maculans*/*V. longisporum* treatments; yield in the 1:1 *L. maculans*/*V. longisporum* treatment was intermediate. Similar trends were observed for ‘CS2000’ (Table 4). The lowest yields were observed in the non-inoculated control and *V. longisporum* only treatment, and the highest was recorded in the *L. maculans* only treatment; yields in the various mixes of *L. maculans* and *V. longisporum* were intermediate (Table 4). Symptoms and signs of Verticillium stripe and blackleg in the greenhouse resembled those described above for the field experiments.

## 3. Discussion

Blackleg is an established disease of canola in Canada, while Verticillium stripe has only recently emerged as a concern in this crop [1,17]. An improved understanding of the impact of blackleg on canola yields, particularly on blackleg-resistant hybrids, as well as of the potential effects of interactions between *L. maculans* and *V. longisporum* on infected crops, is important to determine the need for and effectiveness of different disease management strategies. To our knowledge, this is the first report examining blackleg/Verticillium stripe interactions in canola.

In general, blackleg severity was low in the field experiments to evaluate yield losses caused by this disease, and never exceeded a mean rating of 2.4 for either canola hybrid in either year of the study. This likely reflected the classification of both ‘45H31’ and ‘CS2000’ as blackleg-resistant, although no information regarding the specific resistance genes in these hybrids is available. The *L. maculans* isolates used as inoculum in the field plot experiments were classified as Pathogenicity Group (PG)-2, *sensu* Mengistu [20,21]. The PG classification system is based on the virulence of the blackleg fungus on the differential canola genotypes ‘Westar’, ‘Quinta’ and ‘Glacier’, where PG-2 isolates are avirulent on ‘Quinta’ (carrying the *Rlm1* resistance gene) and ‘Glacier’ (carrying *Rlm2* and *Rlm3*). More recently, *L. maculans* isolates are often classified based on the avirulence (*Avr*) genes they carry, which interact with specific major resistance genes in the host [22]. The isolates used in this experiment were confirmed to carry the *AvrLm4*–*7* and *AvrLm6* avirulence genes, while *AvrLm1* was absent [21]. Although the isolates were not tested for the presence of *AvrLm2* or *AvrLm3*, based on their virulence phenotypes, PG-2 isolates are expected to carry *AvrLm2* (which interacts with *Rlm2*) and *AvrLm3* (which interacts with *Rlm3*) [23]. Most commercial canola hybrids possess the *Rlm3* gene, and PG-2 was predominant in western Canadian *L. maculans* populations for a long time [24,25]. However, additional *Avr* genes beyond *AvrLm2* and *AvrLm3* have been identified in PG-2 [22].

Blackleg development is also influenced by environmental conditions, with humid weather and warm temperatures favoring the disease [26]. This may explain why the mean blackleg severity ratings on both canola hybrids at both sites were higher in 2019 than in 2020. More rainfall in June and July 2019 [27] likely resulted in more severe *L. maculans* infection in the inoculated treatments, leading to more severe disease. In addition, heavy rainfall in May 2020 [27] shortly after inoculation may have flushed the inoculum through the soil, resulting in less disease that year.

In the case of ‘45H31’ and ‘CS2000’, quadratic equations best explained the relationship between blackleg severity and yield. Very mild symptoms (disease severity rating of 1) were associated with an increase in yield relative to plants with no symptoms at all (rating of 0), but as disease severity increased to ≥2, yields decreased dramatically. These results are similar to those reported by Wang et al. [15] in an analysis of moderately resistant canola hybrids. As such, the current study and the earlier report by Wang et al. [15] suggest that quadratic relationships between blackleg severity and yield loss are common in blackleg-resistant and moderately resistant canola. Hwang et al. [14], in contrast, found a linear relationship between disease severity and yield loss in a susceptible canola cultivar, ‘Westar’. It is difficult, however, to draw any conclusions as to whether the level of resistance influences the nature of the relationship between disease severity and yield loss. In the evaluation of 12 commercial canola crops conducted in the current study, the two hybrids (‘DKTF 94CR’ and ‘75-42CR’) exhibited different blackleg severity/yield loss relationships despite both being rated as blackleg-resistant. In ‘DKTF 94CR’, a quadratic relationship was observed, while in ’75-42C’, the relationship was linear. Despite their rating as resistant, moderate levels (2.9 and 2.7) of blackleg were found on both hybrids, indicating that even if there was a similar erosion of resistance in both hosts, the relationship between yield loss and disease severity remained distinct. Nonetheless, at higher blackleg severities (2–5), either model (quadratic or linear) would provide a fairly accurate estimate of yield losses.

In the field experiments examining the interaction between *L. maculans* and *V. longisporum*, symptoms of blackleg were detected in both the control (non-inoculated) and *V. longisporum* inoculum only treatments. These symptoms may have reflected the presence of natural inoculum at the field sites, the spread of asexual pycnidiospores via rain-splash, and/or the fact that wind-borne ascospores of *L. maculans* can travel 5–8 km from the source [28,29]. This was particularly evident in 2020, a year with higher precipitation that was favorable for blackleg, when no strong relationships were found between the amounts of *L. maculans* applied and the severity of blackleg across the treatments. In contrast, 2021 was a very dry year and despite the occurrence of mild symptoms on the control and *V. longisporum* only treatments, blackleg tended to be more severe when more *L. maculans* inoculum was applied. In the case of *V. longisporum*, the situation was reversed, with Verticillium stripe being more severe in 2021 vs. 2020. Verticillium stripe is generally favored by hot and dry conditions, with excess moisture making the disease less problematic [19]. The presence of symptoms of Verticillium stripe on the control and *L. maculans* only treatments suggests the movement of *V. longisporum* microsclerotia via wind-dispersal, rain splash or infested soil [30] from treatments inoculated with the pathogen. Given the increasing prevalence of this fungus in the Prairies [31], it is also possible that there was natural *V. longisporum* inoculum at the field sites, although there was no history of the disease there.

There were no significant differences in seed yields among any of the treatments in each year of the study examining the interaction between *L. maculans* and *V. longisporum*, although yields overall were much greater in 2021 than in 2020. Some of the plots were flooded in 2020 due to heavy rainfall, resulting in yield reductions. In 2021, while there were no statistically significant differences among treatments, numerical differences were observed in the yields between the control and inoculated treatments, with the former being slightly higher in both hybrids at both sites.

Under greenhouse conditions, the emergence of both ‘CS2000’ and ‘45H31’ significantly decreased, relative to the non-inoculated controls, when inoculum of *L. maculans* and/or *V. longisporum* was applied. Barbetti and Khangura [32] reported that blackleg can reduce stand establishment. Similarly, Cui et al. [33] found that percentage emergence was reduced under both low and high concentrations of *V. longisporum* inoculum under greenhouse conditions. Further research is needed to more fully explore the impact of these pathogens on stand establishment, particularly under field conditions.

The non-inoculated control and *V. longisporum* alone treatments had the lowest seed yield for both canola hybrids under greenhouse conditions. The control plants were smaller than the plants in the other treatments as they had higher emergence and hence experienced greater competition. Therefore, single plant seed yield on the non-inoculated control plants was lower than in many of the inoculated treatments. In the treatments receiving *L. maculans* inoculum, blackleg severities ranged from 0.6 to 1.3 across the two hybrids, at which no yield losses are expected according to the empirically derived models from the field experiments. Indeed, yields were highest in these treatments, in comparison not only with the non-inoculated control but also with the *V. longsiporum* only treatments. The results suggest mild Verticillium stripe infection might cause greater yield losses compared with mild blackleg infection, but this may not be the case at higher disease severities.

Blackleg is generally diagnosed on canola based on the discoloration of cross-sections of the lower stem, but Verticillium stripe also causes cross-section discoloration. Growers and agronomists who are not familiar with Verticillium stripe might have difficulties identifying this disease, given its relatively novel emergence in Canada [34]. This study demonstrated that longitudinal sections could help to differentiate blackleg and Verticillium stripe more readily on canola, enabling more accurate diagnoses and disease monitoring in the field.

Plant pathogens can be affected by the quantity and/or quality of shared resources, leading to resource-mediated interactions between different pathogens [35]. While, to our knowledge, there have been no previous studies on the interaction between *L. maculans* and *V. longsiporum*, Toscano-Underwood et al. [36] examined the co-existence of two closely related species, *L. maculans* and *L. biglobosa*. Epidemiological differences between these pathogens resulted in stable co-existence [36]. Another study examined two foliar wheat pathogens *Puccinia triticina* (leaf rust) and *Pyrenophora tritici-repentis* (tan spot) under greenhouse conditions and found that infection by *P. graminis* facilitated the development of tan spot [37]. In this study, under both field and greenhouse conditions and on both canola hybrids, blackleg severities were generally higher when *L. maculans* was co-inoculated with *V. longisporum* than when *L. maculans* was applied alone. Microsclerotia of *V. longisporum* germinate and the fungus enters the plant vascular system via formation of hyphae [18], while *L. maculans* colonizes the intercellular spaces and also reaches the vascular tissue [2]. Co-infection of the vascular tissue by both fungi could facilitate nutrient release from degraded host tissues and promote the growth of *L. maculans* even on blackleg resistant or moderately resistant hybrids. The destruction of the stem cortex caused by *L. maculans* could also facilitate microsclerotium formation by *V. longisporum* at the later plant stages. Regardless of the exact mechanism, the results suggest a synergistic effect, with the presence of both pathogens resulting in more severe disease overall.

## 4. Materials and Methods

### 4.1. Inoculum Preparation

The isolates of *L. maculans* (11C78103, 11C78301, 11S54041, 11P125232, and 11L999014) used as inoculum in the field plot and greenhouse experiments were originally characterized by Rong et al. [20], while the isolate of *V. longisporum* (Vlss43) was recovered from diseased *B. napus* tissues collected near Edmonton, Alberta. Grain inoculum of both fungi was prepared based on Hwang et al. [38]. Briefly, cultures of *L. maculans* were grown in Petri dishes on V8 medium (composition per liter: 850 mL distilled water, 150 mL V8^®^ Original Vegetable Juice (Campbell Soup Company, Camden, NJ, USA), 1.5 g CaCO3, 15.0 g agar) and then incubated for 21 d at room temperature under fluorescent lighting to encourage pycnidiospore production. Cultures of *V. longisporum* were grown in Petri dishes on potato dextrose agar (PDA; 26 g PDA powder, 850 mL distilled water) and then incubated in darkness at room temperature for 21 d for conidial production. The colonies of *L. maculans* and *V. longisporum* were cut into small pieces and mixed with sterilized, water-soaked wheat grain (900 mL grain per one culture of *L. maculans* or *V. longisporum*) in separate autoclave bags. The inoculated grain was incubated at room temperature for 28 d and then dried at 35 °C for 2 d. After drying, the inoculated grain was ground in a grain mill and passed through a mesh sieve that was 2 mm in diameter.

### 4.2. Field Experiments for Blackleg Yield Losses

Field experiments to evaluate the relationship between blackleg severity and yield were conducted over 2 years (2019 and 2020) at the Crop Diversification Centre-North, Edmonton, AB, Canada (53°39′ N, 113°22′ W). Two blackleg-resistant canola hybrids, ‘45H31’ and ‘CS2000’, were included in the experiments. Treatments were arranged in a split-plot design with four replicates. Each plot consisted of four rows, 6 m in length and 1.5 m in width, with a 0.25-meter spacing between the rows. Adjacent plots were separated by a m buffer zone of 1 m, with 2 m between replicates. Each row was seeded with 0.7 g of canola using a push seeder. The grain inoculum was applied (200 mL/row) at seeding by placing it in the seeder together with the seeds. No grain inoculum was included in the control treatments. The plots were seeded on 27 May 2019 and 17 May 2020.

To assess the impact of blackleg on canola yield, all plants within a 1 m^2^ area in the center of each plot were carefully excavated from the soil with a shovel at maturity (8 October 2019 and 1 October 2020) and placed in paper bags. The remainder of each plot was harvested with a small plot combine using a straight cut header, and the seed was weighed to determine overall yield. The average blackleg severity per 1 m^2^ plot area was assessed on a 0–5 scale as described below, and stem cross-sections were examined by cutting the plant at the soil line. The number of pods and seed yield were individually recorded for each plant. The pods from each plant were threshed manually, and the seeds were cleaned and weighed.

### 4.3. Blackleg Yield Losses in Commercial Crops

To evaluate yield losses due to blackleg in commercial fields, nine canola crops in the County of Wetaskiwin and three crops in the County of Lacombe, located in central Alberta, were sampled at maturity on 12 September 2019. These crops were selected for sampling based on the occurrence of symptoms of blackleg. Briefly, entire plants were dug out from the soil within a 1 m^2^ area at each of 5 locations along the arms of a ‘W’ sampling pattern, with 100 canola plants collected per field [39]. The plants were placed in paper bags for subsequent assessment in the laboratory. Each plant was rated for blackleg severity on the 0–5 scale described below, and pod number and seed yield were recorded. Pods were threshed manually, and the seeds were cleaned and weighed. Six of the fields sampled in Lacombe County were sown to the canola hybrid ‘DKTF 94CR’, and three were sown to ‘75-42CR’; both hybrids are rated as blackleg resistant. The three fields in the County of Lacombe were all sown to the hybrid ‘DKTF 94CR’.

### 4.4. Field Experiments for Blackleg and Verticillium Stripe Interactions

Field trials to evaluate the effect of blackleg/Verticillium stripe interactions on canola yield were conducted over 2 years (2020 and 2021) at two sites located at the Crop Diversification Centre-North. The canola hybrids ‘45H31’and ‘CS2000’were included in the experiments, which were arranged in a split-plot design with four replicates. Each plot consisted of four rows, 6 m in length and 1.5 m in width with 0.25-meter spacing between the rows. Adjacent plots were separated by a buffer zone of 1 m, with 2 m between replicates. Each row was seeded with 0.7 g of canola as described above. Grain inoculum of *L. maculans* and *V. longisporum* was applied at various ratios in the different treatments: *L. maculans* grain inoculum applied at 200 mL/row; *V. longisporum* grain inoculum at 200 mL/row; a 3:1 mix of *L. maculans* (150 mL/row) and *V. longisporum* (50 mL/row); a 1:1 mix of *L. maculans* (100 mL/row) and *V. longisporum* (100 mL/row); and a 1:3 mix of *L. maculans* (50 mL/row) and *V. longisporum* (150 mL/row) inoculum. Control treatments did not receive any inoculum. The grain inoculum was applied at seeding by placing it in the seeder as above. Experiments were seeded on 17 May 2020 and 18 May 2021.

To assess the impact of blackleg and Verticillium stripe interactions, 15 plants from each plot were carefully collected with a shovel and placed in paper bags. The remainder of each plot was harvested with a small plot combine using a straight cut header, and the seed was weighed to determine overall yield. The percentage yield reduction was calculated relative to non-inoculated controls. Experiments were harvested at maturity on 13 Oct. 2020 and 22 Sept. 2021. The plants were rated for blackleg severity on a 0–5 scale and for Verticillium stripe severity on a 0–4 rating scale as described below. Plant samples were visually examined for the presence of pycnidia of *L. maculans* and microsclerotia of *V. longisporum*. Horizontal and vertical sections of the stems were made with a bypass pruner for comparison of blackleg and Verticillium stripe symptoms.

### 4.5. Greenhouse Experiments for Blackleg and Verticillium Stripe Interactions

Greenhouse experiments were conducted with the canola cultivars ‘45H31’ and ‘CS2000’. The experiments were arranged in a split-plot design with four replicates using plastic containers (40.9 × 28.2 × 15.0 cm) filled with Sunshine^®^ mix #4 potting medium. Two rows were seeded per container, at a rate of 20 seeds per row, and grain inoculum was placed along with the seeds. Inoculation treatments included: *L. maculans* grain inoculum applied at 20 mL/row; *V. longisporum* grain inoculum applied at 20 mL/row; a 3:1 mix of *L. maculans* (15 mL/row) and *V. longisporum* (5 mL/row); a 1:1 mix of *L. maculans* (10 mL/row) and *V. longisporum* (10 mL/row); and a 1:3 mix of *L. maculans* (5 mL/row) and *V. longisporum* (15 mL/row). Control treatments were not inoculated. The experiment was repeated. All the plants in each container were rated for blackleg severity on a 0–5 scale and Verticillium stripe severity on a 0–4 scale as described below. Horizontal and vertical sections were made for identification of blackleg and Verticillium stripe. Emergence counts were taken 14 days after seeding. Seed yields were weighed and recorded.

### 4.6. Disease Assessments

Plants were rated for blackleg severity on a 0–5 scale, where 0 = no infection; 1 = lesion area < 25% of the cross-section area of the crown; 2 = lesion area 25–50% of the cross-section area of the crown; 3 = lesion area 51–75% of the cross-section area crown; 4 = lesion area 76–100% of the cross-section area of the crown; and 5 = plant dead [40]. Verticillium stripe severity was assessed on a 0–4 scale based on the amount of fungal microsclerotia on the entire plant, where: 0 = healthy plants with no microsclerotia visible; 1 = slight colonization by microsclerotia < 25%; 2 = moderate colonization by microsclerotia < 75%; 3 = extensive colonization by microsclerotia > 75%; 4 = severe colonization by microsclerotia and peeling of the stem epidermis.

### 4.7. Statistical Analysis

Statistical analysis was conducted using R: A Language and Environment for Statistical Computing (R Core Team, R Foundation for Statistical Computing, Vienna, Australia, 2013). To establish the relationship between blackleg severity and pod number and seed yield, regression analysis was performed. Akaike information and Bayesian information criteria were used for selection of the best model for the data. Adjusted R^2^ values and the F test were used to examine compatibility of the regression. Residual data were tested for normality with the Shapiro–Wilk test in R shapiro.test stats. Regression equations were generated to evaluate the losses in pod number and seed yield with increasing disease severity. The yield of plants with no blackleg symptoms was used as a point estimate, with different yield data points at each disease severity transformed into yield percentages relative to canola yield with no disease. Regression analysis was performed to estimate yield loss percentage per unit increase in disease severity. To examine blackleg and Verticillium stripe interactions, canola hybrid was considered as a fixed effect, and replication and site-year and their interaction as random effects. Analysis of variance was performed. Least significant difference comparisons were used to determine whether disease severity and seed yield differed among concentrations.

## 5. Conclusions

In this study, the relationship between blackleg severity and yield components was explained best by second-degree quadratic equations in most canola hybrids examined, although a linear relationship was observed for one variety sampled in commercial fields. Under natural field conditions, however, multiple plant pathogens may occur together. The recent identification of Verticillium stripe in Canada is of particular concern for canola growers, and the disease may be found in conjunction with blackleg in some fields. When *L. maculans* and *V. longisporum* were inoculated together in field and greenhouse experiments, blackleg severity and yield losses increased relative to when *L. maculans* was applied on its own. The severity of Verticillium stripe also tended to increase, suggesting a synergistic effect between the pathogens. Under low inoculum pressure, *V. longisporum* caused more severe yield losses than blackleg. The two diseases could be readily distinguished by longitudinal sections of the lower stems, which will facilitate surveillance activities and identification by non-expert personnel. The co-occurrence of blackleg and Verticillium stripe on canola represents another challenge to Canadian canola production and will require the development of proactive disease management strategies.

## Figures and Tables

**Figure 1 plants-12-00434-f001:**
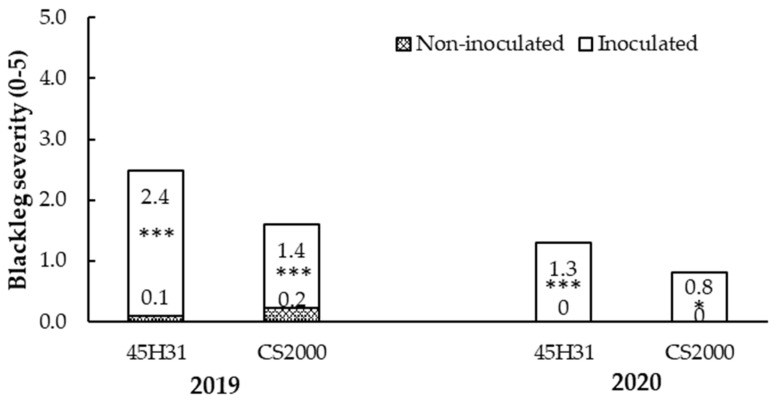
Mean blackleg disease severity on the canola hybrids ‘45H31’ and ‘CS2000’ under field conditions in *Leptosphaeria maculans*-inoculated and non-inoculated treatments. Data were collected over two years (2019 and 2020) at two sites in Edmonton, AB, Canada. Blackleg severity was assessed on a 0–5 scale, where 0 = no disease and 5 = death of the plant. Significant differences between inoculated and non-inoculated plots of each hybrid in each year are indicated with asterisks: *, *p* < 0.05; ***, *p* < 0.001.

**Figure 2 plants-12-00434-f002:**
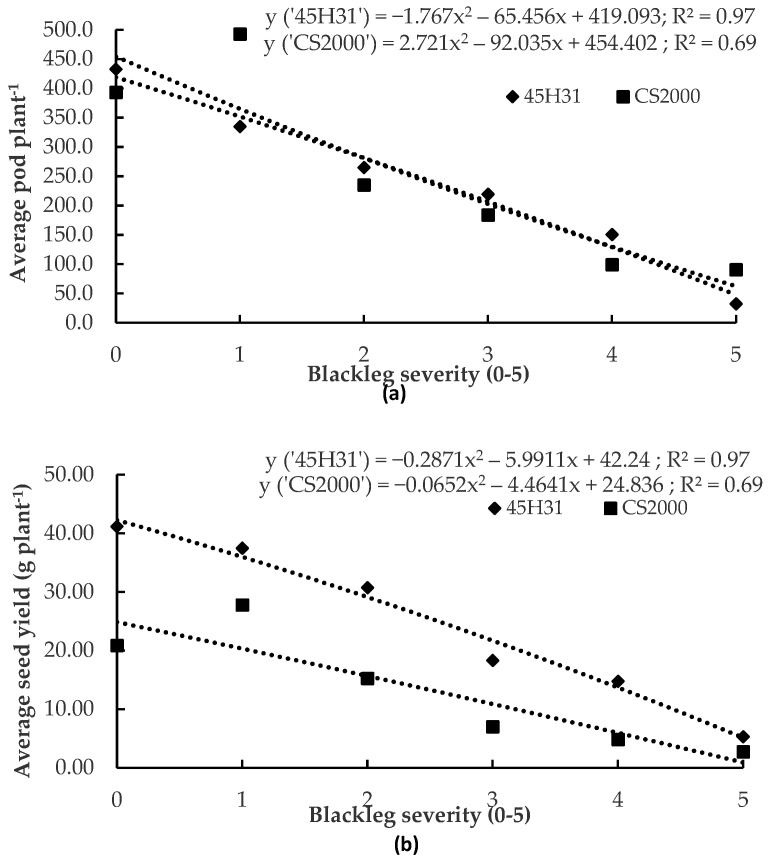
Relationship between blackleg severity and pods per plant (**a**) or seed yield per plant (**b**) in the canola hybrids ‘45H31’ and ‘CS2000’ under field conditions. Data were collected over two years (2019 and 2020) at two sites in Edmonton, AB, Canada. Each point represents the mean of four replications × two site-years. Blackleg severity was assessed on a 0–5 scale, where 0 = no disease and 5 = death of the plant.

**Figure 3 plants-12-00434-f003:**
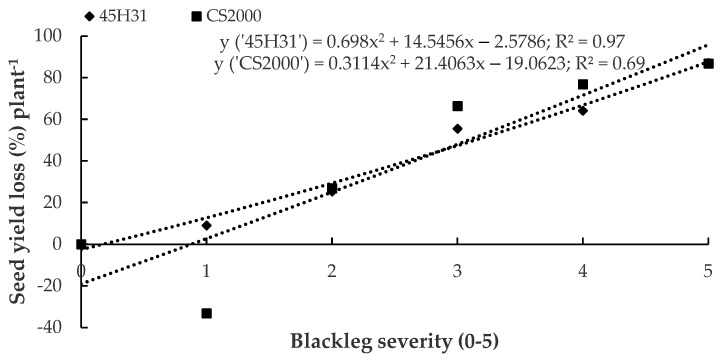
Relationship between blackleg severity and yield loss in the canola hybrids ‘45H31’ and ‘CS2000’ under field conditions. Data were collected over two years (2019 and 2020) at two sites in Edmonton, AB, Canada. The yield loss data were estimated using the y-intercept in the equation averaged over four site-years. The data points were transformed into the percentage of the maximum yield.

**Figure 4 plants-12-00434-f004:**
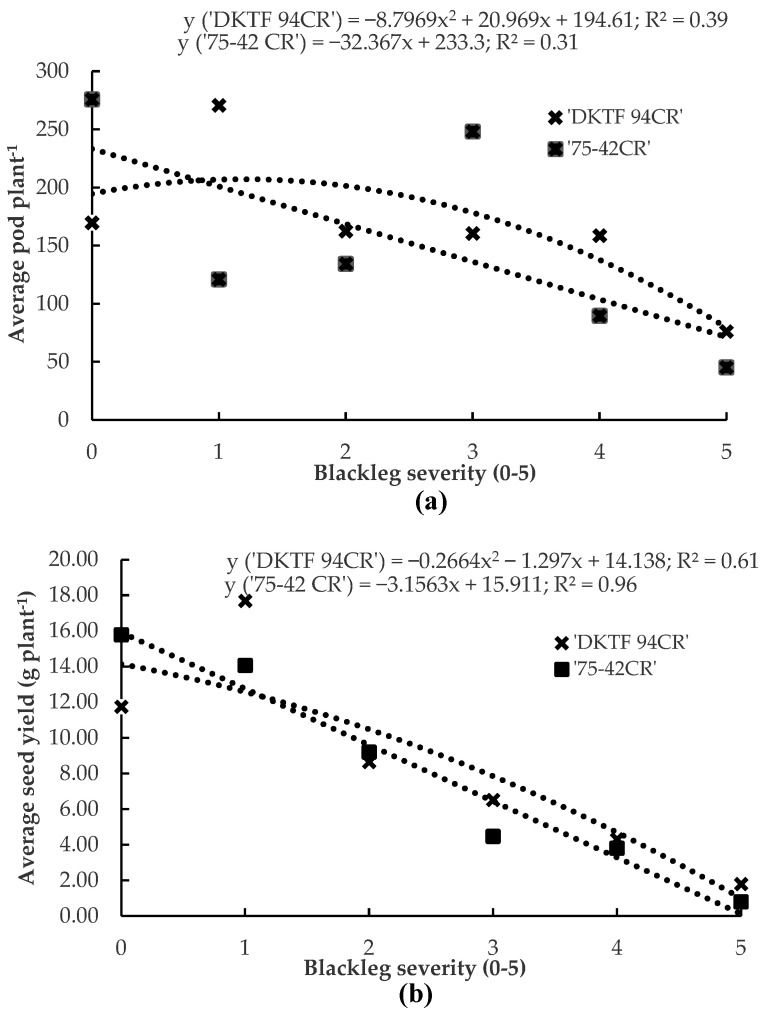
Relationship between blackleg severity and pods per plant (**a**) and seed yield per plant (**b**) in the canola hybrids ‘DKTF 94CR’ and ‘75-42CR’ sampled in 12 commercial fields around Lacombe and Wetaskiwin, AB, Canada, in 2019. Each point represents the mean of all fields planted to the same canola hybrid. Blackleg severity was assessed on a 0–5 scale, where 0 = no disease and 5 = death of the plant.

**Figure 5 plants-12-00434-f005:**
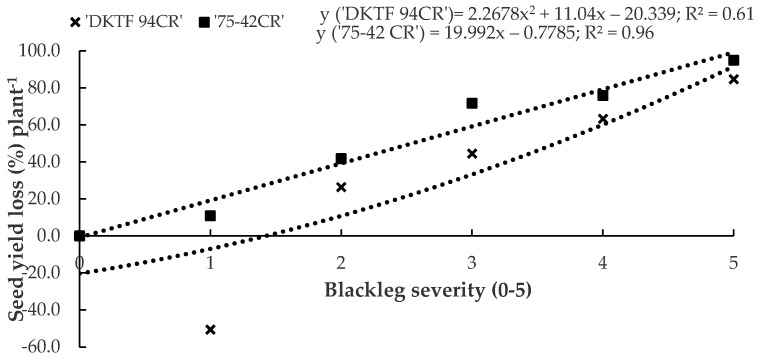
Relationship between blackleg severity and yield loss in the canola hybrids ‘DKTF 94C’ and ‘75-42CR’ sampled in 12 commercial fields around Lacombe and Wetaskiwin, AB, Canada, in 2019. The yield loss data were estimated using the y-intercept in the equation averaged over 12 commercial fields. The data points were transformed into the percentage of the maximum yield.

**Figure 6 plants-12-00434-f006:**
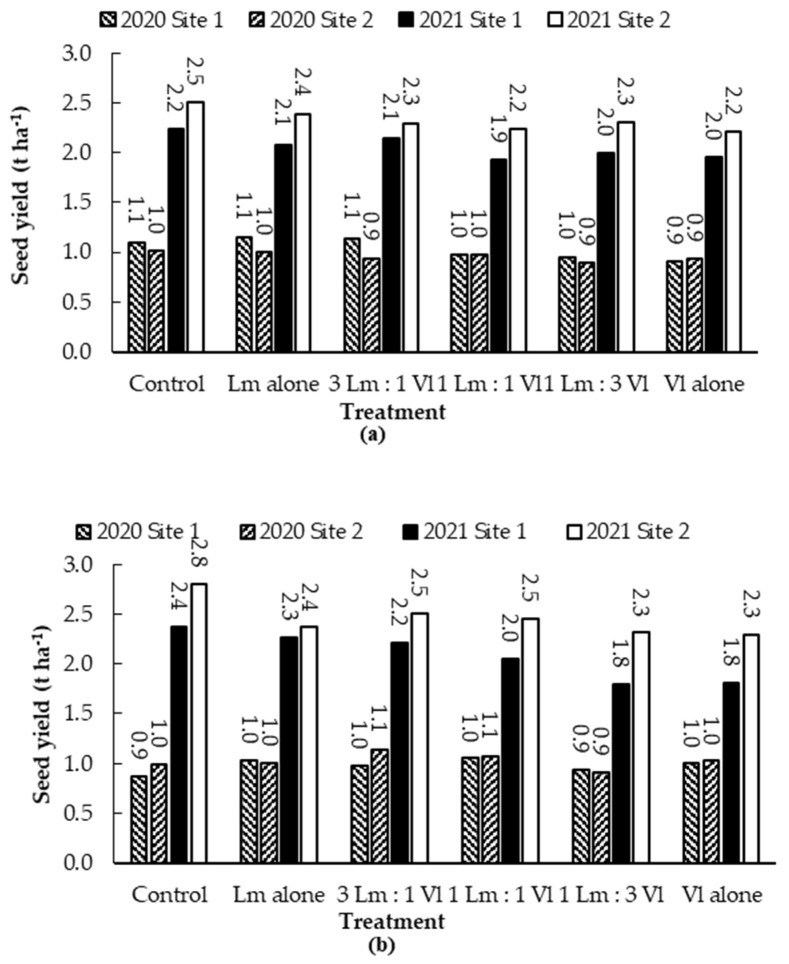
Mean seed yield of the canola hybrids ‘45H31’ (**a**) and ‘CS2000’ (**b**) under field conditions. Data were collected over two years (2020 and 2021) in Edmonton, AB, Canada, following inoculation with *Leptosphaeria maculans* (*Lm*) and *Verticillium longisporum* (*Vl*) alone or in various combinations (3:1, 1:1, 1:3). Values represent the mean of four replications for each year. Mean seed yields were not significantly different according to the Tukey-Kramer test (*p* > 0.05) among any of the treatments.

**Figure 7 plants-12-00434-f007:**
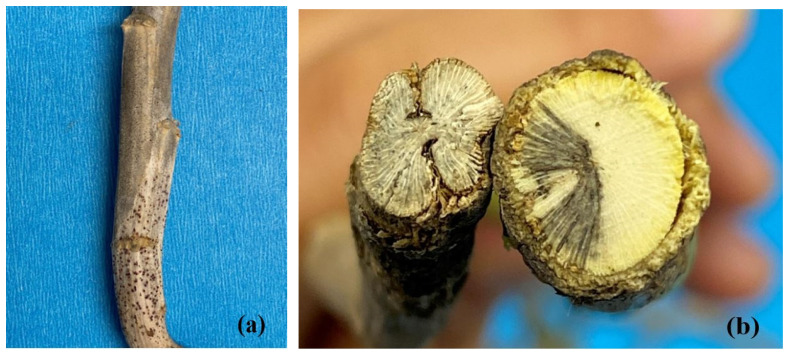
Pycnidia of *Leptosphaeria maculans* (lower portion of stem) and microsclerotia of *Verticillium longisporum* (upper portion) occurring on the same canola stem (**a**). Cross-sections of canola stems showing the discoloration caused by Verticillium stripe (left) and blackleg (right) (**b**).

**Figure 8 plants-12-00434-f008:**
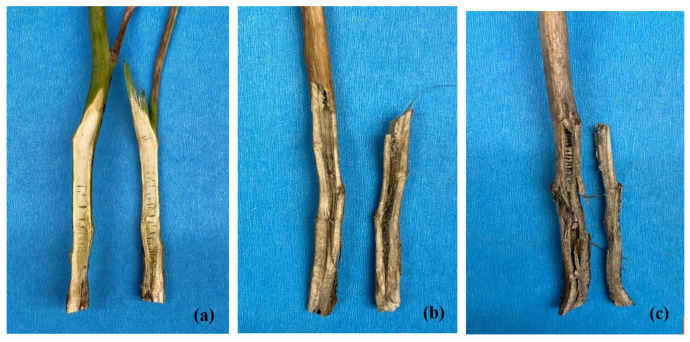
Longitudinal sections of canola stems infected by *Leptosphaeria maculans* (**a**), *Verticillium longisporum* (**b**), and both pathogens (**c**).

**Table 1 plants-12-00434-t001:** Yield (t ha^−1^) of canola hybrids ‘45H31’ and ‘CS2000’ in *Leptosphaeria maculans*-inoculated and non-inoculated treatments in field trials near Edmonton, AB, Canada, in 2019 and 2020.

Year	Treatment	‘45H31’	‘CS2000’
2019	Non-inoculated	3.47 a ^1^	2.72 a
	Inoculated	2.18 b	1.87 b
2020	Non-inoculated	1.05 A	0.88 A
	Inoculated	1.00 A	0.82 A

^1^ Data are the means of four replications; means in a column followed by the same letter are not significantly (*p* ≤ 0.05) different for each year according to the Tukey-Kramer test.

**Table 2 plants-12-00434-t002:** Blackleg severity (0–5) on the canola hybrids ‘45H31’ and ‘CS2000’ following inoculation with *Leptosphaeria maculans* (*Lm*) and/or *Verticillium longisporum* (*Vl*) alone and in various combinations under field conditions.

Treatment ^1^	2020				2021			
	Site 1		Site 2		Site 1		Site 2	
	‘45H31’	‘CS2000’	‘45H31’	‘CS2000’	‘45H31’	‘CS2000’	‘45H31’	‘CS2000’
Control	0.4 d ^2^	0.2 B	0.1 b	0.1 B	0.5 c	0.2 C	0.6 bc	0.0 C
*Lm* alone	1.0 bcd	0.9 A	0.7 ab	0.8 A	1.5 a	1.1 A	1.0 a	1.2 A
3 *Lm*: 1 *Vl*	1.3 abc	0.9 A	1.2 a	1.0 A	1.0 ab	0.6 B	0.4 bc	0.4 BC
1 *Lm*: 1 *Vl*	1.6 a	1.0 A	1.5 a	0.8 A	0.6 ab	0.6 BC	0.3 b	0.6 AB
1 *Lm*: 3 *Vl*	1.1 cd	0.9 A	0.9 ab	0.9 A	0.3 bc	0.5 B	0.4 bc	0.3 BC
*Vl* alone	1.5 ab	1.3 A	1.0 ab	1.2 A	0.7 c	0.3 BC	0.6 c	0.4 BC

^1^ *Lm* alone = *Lm* applied at 200 mL inoculum/row; 3 *Lm*: 1 *Vl* = 3:1 mix of *Lm* (150 mL/row) and *Vl* (50 mL/row); 1 *Lm*: 1 *Vl* = 1:1 mix of *Lm* (100 mL/row) and *Vl* (100 mL/row); 1 *Lm*: 3 *Vl* = 1:3 mix of *Lm* (50 mL/row) and *Vl* (150 mL/row); *Vl* alone = *Vl* applied at 200 mL/row inoculum. ^2^ Data were collected over four site-years in Edmonton, AB, Canada and are the means of four replications. Means in a column followed by the same letter are not significantly (*p* ≤ 0.05) different according to the Tukey-Kramer test.

**Table 3 plants-12-00434-t003:** Verticillium stripe severity (0–4) on the canola hybrids ‘45H31’ and ‘CS2000’ following inoculation with *Leptosphaeria maculans* (*Lm*) and/or *Verticillium longisporum* (*Vl*) alone and in various combinations under field conditions.

Treatment ^1^	2020				2021			
	Site 1		Site 2		Site 1		Site 2	
	‘45H31’	‘CS2000’	‘45H31’	‘CS2000’	‘45H31’	‘CS2000’	‘45H31’	‘CS2000’
Control	0.0 c ^2^	0.1 C	0.0 a	0.2 B	0.5 b	0.2 C	0.6 c	0.0 B
*Lm* alone	0.3 b	0.3 BC	0.0 a	0.2 B	1.1 b	0.4 C	1.1 c	0.2 B
3 *Lm*: 1 *Vl*	0.4 ab	0.3 ABC	0.4 a	0.3 AB	1.0 a	0.5 BC	2.2 b	1.5 A
1 *Lm*: 1 *Vl*	0.4 ab	0.6 AB	0.2 a	0.2 AB	1.0 a	0.8 ABC	1.7 b	1.8 A
1 *Lm*: 3 *Vl*	0.2 bc	0.5 ABC	0.2 a	0.2 AB	1.1 a	1.2 AB	1.8 a	1.6 A
*Vl* alone	0.5 a	0.7 A	0.3 a	0.6 A	1.2 a	1.6 A	1.9 a	2.0 A

^1^ *Lm* alone = Lm applied at 200 mL inoculum/row; 3 *Lm*: 1 *Vl* = 3:1 mix of *Lm* (150 mL/row) and *Vl* (50 mL/row); 1 *Lm*: 1 *Vl* = 1:1 mix of *Lm* (100 mL/row) and *Vl* (100 mL/row); 1 *Lm*: 3 *Vl* = 1:3 mix of *Lm* (50 mL/row) and *Vl* (150 mL/row); *Vl* alone = *Vl* applied at 200 mL/row inoculum. ^2^ Data were collected over four site-years in Edmonton, AB, Canada and are the means of four replications. Means in a column followed by the same letter are not significantly (*p* ≤ 0.05) different according to the Tukey-Kramer test.

**Table 4 plants-12-00434-t004:** Seedling emergence, blackleg severity, Verticillium stripe severity and seed yield of the canola hybrids ‘45H31’ and ‘CS2000’ following inoculation with *Leptosphaeria maculans* (*Lm*) and/or *Verticillium longisporum* (*Vl*) alone and in various combinations under greenhouse conditions.

Treatment ^1^	Emergence (%) ^2^		Blackleg Severity (0–5)		Verticillium stripe Severity (0–4)		Yield (g plant^−1^)	
	‘45H31’	‘CS2000’	‘45H31’	‘CS2000’	‘45H31’	‘CS2000’	‘45H31’	‘CS2000’
Control	94.7 a	95.0 A	0.0 d	0.0 C	0.0 c	0.0 B	1.5 c	1.2 B
*Lm* alone	41.6 c	52.5 B	0.9 b	0.8 AB	0.0 c	0.0 B	3.9 a	2.3 A
3 *Lm*: 1 *Vl*	42.2 c	52.5 B	1.3 a	1.0 A	1.0 b	1.6 A	3.8 a	2.0 AB
1 *Lm*: 1 *Vl*	54.7 b	62.2 B	0.9 bc	0.8 AB	1.0 b	1.7 A	3.4 ab	1.8 AB
1 *Lm*: 3 *Vl*	57.2 b	63.8 B	0.6 c	0.6 B	1.3 b	1.8 A	2.7 b	1.7 AB
*Vl* alone	55.6 b	51.9 B	0.0 d	0.0 C	1.9 a	1.9 A	1.6 c	1.4 B

^1^ *Lm* alone = *Lm* applied at 20 mL inoculum/row; 3 *Lm*: 1 *Vl* = 3:1 mix of *Lm* (15 mL/row) and *Vl* (5 mL/row); 1 *Lm*: 1 *Vl* = 1:1 mix of *Lm* (10 mL/row) and *Vl* (10 mL/row); 1 *Lm*: 3 *Vl* = 1:3 mix of *Lm* (5 mL/row) and *Vl* (15 mL/row); *Vl* alone = *Vl* applied at 20 mL/row inoculum. ^2^ Data are the means of four replications in each of two repeats of the experiment, which were combined as they were not significantly different (*p* > 0.05); means in a column followed by the same letter are not significantly different according to the Tukey-Kramer test (*p* ≤ 0.05).

## Data Availability

Data are available from the corresponding author upon reasonable request.

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
