# Peer review of "Blackleg Yield Losses and Interactions with Verticillium Stripe in Canola (Brassica napus) in Canada"

_plants, 2023, doi:10.3390/plants12030434_

Round 1

Reviewer 1 Report

In addition to being a significant source of revenue for 43,000 Canadian farmers, canola has grown to be one of Canada's most valued agricultural exports. Canola is currently responsible for more than 207,000 jobs in Canada, and this number is expected to rise. However, a number of biotic and abiotic variables restrict the amount of canola produced in Canada. In vulnerable types, the deadly canola disease blackleg can result in significant output losses. Blackleg epidemics can cause yield losses of 30% to 50%. Verticillium stripe, a disease brought on by Verticillium longisporum, has also just come to light as a possible hazard to canola in Canada. Given that blackleg and Verticillium stripe are now present on canola in Canada, it's probable that some L. maculans infections are mistaken for V. longisporum infections. Therefore, it's crucial to determine whether and how canola co-infection influences yield losses brought on by V. longisporum and L. maculans. The authors choose the appropriate subject to work on with specific goals after taking into account the economic significance of both of these diseases for the Canadian canola industry. Work has been done in both artificial and natural settings, and the experimental design is acceptable. The study's findings, which show a synergistic relationship between the viruses, are highly intriguing. Additionally, they contend that the co-occurrence of Verticillium stripe and blackleg on canola poses a new obstacle to Canadian canola production and necessitates the creation of pro-active disease management techniques. Overall, the findings of this study are original and highly fascinating; they may be applied in the development of management plans to prevent certain illnesses in canola. Additionally, the work is beautifully written and really well organized. The paper's opening section has enough recent data to support the study's goals. Tables and figures are used to portray the findings in an attractive manner. The study's findings are supported by the conclusion, which is written extremely simply. I must praise the authors of the study on their good job overall and urge the editor-in-chief to accept this manuscript for publication in plants.

Reviewer 2 Report

Dear Authors,

The title of the article sounds interesting. Also, performed analyses  look interesting. Article focuses on the important issue of yield losses in canola

The experiments are reproducible. Results presented in a clear authoritative reliable form.

Presented conclusions are the result of the analysis carried out. Comprehensive discussion.

In my opinion, the manuscript is almost ready to public.

However:

1.    4.5. Greenhouse experiments for blackleg and Verticillium stripe interactions

- There is information about replication- the experiment was repeated, how many times?

2.  Record of   1-m2  , why there is a line, almost in all manuscript

3. Please read the MS once more and correct any minor shortcomings, e.g. punctuation, etc.
